# Association of helicobacter pylori infection with lipid metabolism and 10-year cardiovascular risk in diabetes mellitus: A cross-sectional study

**Yuexi Li**◉, **Xiaoqin Liu***, **Qing Li, Peng Zhou, Qian Chen, Bolan Jiang, Taiju Zhu**

Health Management Center, Deyang People's Hospital, Deyang, Sichuan, China

◉ These authors contributed equally to this work and should be considered co-first authors.
* 66823572@qq.com

## Abstract

### Background

Previous studies have shown that Helicobacter pylori infection is not only a risk factor for gastrointestinal diseases but also associated with various non-digestive conditions. This study aimed to investigate the effect of Helicobacter pylori infection on the risk of lipid metabolism disorders and cardiovascular disease in individuals with diabetes mellitus.

### Methods

This cross-sectional study was conducted at a health examination center. Data from life questionnaires, laboratory tests, the carbon-13 urea breath test, and the Framingham Risk Score were collected from 266 patients with diabetes. All participants were categorized into Helicobacter pylori-uninfected and Helicobacter pylori-infected groups based on the carbon-13 urea breath test results. Differences in lipid levels, Framingham Risk Score, and cardiovascular disease risk were compared between the two groups. A logistic regression model was applied to analyze whether Helicobacter pylori infection is an independent risk factor for dyslipidemia in patients with diabetes.

### Results

Total cholesterol and low-density lipoprotein cholesterol levels were higher in the Helicobacter pylori-infected group than in the uninfected group, and high-density lipoprotein cholesterol levels were lower in the infected group (both P < 0.05). There was no statistically significant difference in triglyceride levels between the two groups. Regression analysis showed that Helicobacter pylori infection was an independent risk factor for dyslipidemia in patients with diabetes (P < 0.05). The Framingham Risk Score and 10-year cardiovascular disease risk were higher in the Helicobacter pylori-infected group compared with the uninfected group (P < 0.001).

**Data availability statement:** All relevant data are within the manuscript and its Supporting Information files.

**Funding:** Our study was supported by the Sichuan Medical and Health Care Promotion Association (Project No. KY2022SJ0100).

**Competing interests:** The authors have declared that no competing interests exist.

## Conclusion

Helicobacter pylori infection is associated with dyslipidemia and may contribute to an increased risk of cardiovascular disease in individuals with diabetes.

## Introduction

*Helicobacter pylori* (HP) is a gram-negative bacterium that colonizes the gastric mucosa and is a well-established cause of digestive diseases, including peptic ulcers [1] and gastric cancer [2,3]. Beyond its gastrointestinal effects, HP has been implicated in a range of non-gastrointestinal (non-GI) diseases, such as Parkinson's disease [4], Alzheimer's disease [5], stroke [6], atherosclerosis [7], and non-alcoholic fatty liver disease [8]. While these associations suggest a broader systemic impact of HP, the precise nature of its role—whether positive, negative, or neutral—remains an area of active investigation. Emerging evidence further highlights HP's potential influence on systemic inflammation and metabolic disturbances, including its association with diabetes mellitus and cardiovascular disease [9,10].

Dyslipidemia, characterized by elevated levels of total cholesterol (TC) [11] and low-density lipoprotein cholesterol (LDL-C) [12], is a well-known risk factor for cardiovascular diseases such as coronary artery disease. Low levels of high-density lipoprotein cholesterol (HDL-C) further exacerbate the risk of atherosclerosis [13] and coronary heart disease [14], with studies demonstrating a negative correlation between HDL-C levels and the severity of coronary artery disease [15]. Recent studies suggest that HP infection may influence lipid metabolism, contributing to dyslipidemia and its associated cardiovascular risks [16,17]. For instance, HP infection has been associated with increased LDL-C levels and reduced HDL-C levels, which are critical contributors to cardiovascular risk [16]. Moreover, HP eradication has been reported to improve lipid profiles in patients with diabetes, suggesting a potential therapeutic benefit [18].

Diabetes mellitus is a condition marked by chronic metabolic disturbances, including dyslipidemia, that significantly elevate the risk of cardiovascular complications. Studies have demonstrated that individuals with diabetes not only have a higher prevalence of HP infection but also face lower eradication rates, suggesting a bidirectional relationship between these conditions [19,20]. Additionally, HP infection may exacerbate systemic inflammation and oxidative stress in diabetic individuals, further contributing to dyslipidemia and cardiovascular risks [9,21]. Despite these findings, limited attention has been given to the specific impact of HP infection on lipid metabolism and cardiovascular risk within diabetic populations. Addressing this gap is crucial, as understanding the interplay between HP infection and metabolic health in diabetes could inform targeted interventions to mitigate cardiovascular risk.

Therefore, this study aims to explore whether HP infection affects lipid metabolism and increases the risk of cardiovascular disease in patients with diabetes. By investigating this relationship, we hope to contribute to a more comprehensive understanding of the systemic effects of HP infection and its implications for the management of diabetes-related complications. To achieve this, we conducted a cross-sectional study using retrospective data from patients with diabetes who underwent health checkups at the Health Management Center of Deyang People's Hospital. Anthropometric measurements, laboratory tests, and lifestyle factors were collected to comprehensively assess lipid metabolism and cardiovascular risk. HP infection was diagnosed using the 13C-urea breath test, and the Framingham Risk Score was employed to estimate 10-year cardiovascular disease (CVD) risk. Statistical analyses, including univariate and multivariate logistic regression, were used to identify independent predictors of lipid metabolism abnormalities and assess the role of HP infection in this context.

## Materials and methods

### Study population

The study was a cross-sectional survey conducted at the Health Management Center of Deyang People's Hospital. Data were obtained from January 1 to May 31, 2024, from archived medical examinations of patients with diabetes who underwent health checkups at the Health Management Center. The study was carried out in accordance with the Declaration of Helsinki and was approved by the Ethics Committee of Deyang People's Hospital (Ethical review approval number: LWH-OP-006-A04-V2.0). The Ethics Committee waived the requirement for informed consent because the study was retrospective, utilizing existing medical examination records of clients. Data were de-identified before analysis to ensure the confidentiality and privacy of participants, in compliance with ethical research standards and data protection regulations.

The study focused on patients diagnosed with Type 2 diabetes mellitus who met the inclusion criteria of being willing to participate fully and having reliable laboratory results. To ensure robust and complete data, exclusions were made for individuals who declined certain measurements, had incomplete questionnaires, or were diagnosed with specific conditions such as tumors, liver diseases (e.g., hepatitis, cirrhosis, or schistosomiasis-related liver disease), or unreliable lab results. Unreliable laboratory test results were defined as those with hemolyzed samples during analysis. These cases were identified and documented by trained laboratory technicians following standard operating procedures. We excluded participants with tumors and hepatitis to minimize confounding effects, as these conditions are known to independently influence systemic inflammation and metabolic parameters, potentially biasing the study outcomes. The above criteria helped establish a well-defined and representative sample of eligible participants, as detailed in Fig 1.

### Anthropometric, biochemical, and lifestyle data collection

For each participant, age and gender were recorded, height, weight, and waist circumference were taken. The blood pressure was measured three times in the sitting position, and the mean value was taken as the final blood pressure. Body mass index (BMI) is calculated as weight

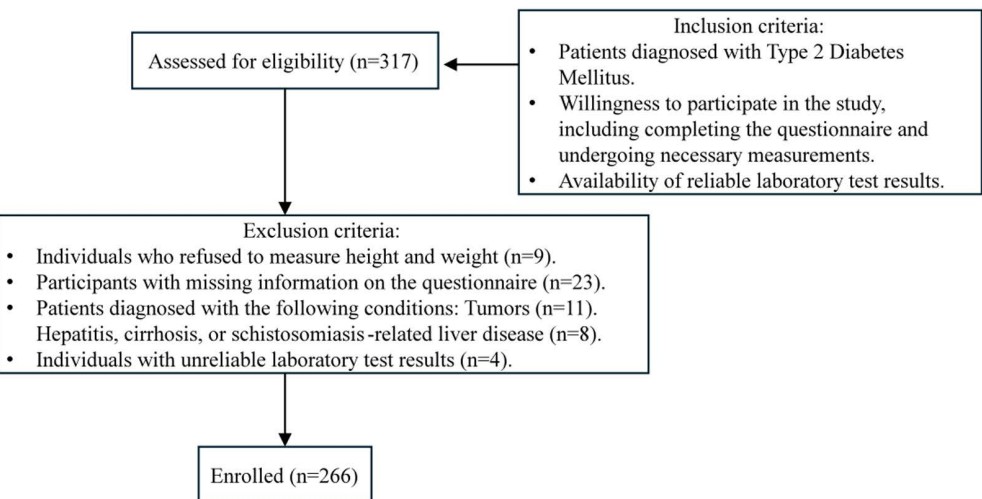

**Fig 1. Graphical flowchart for the participant recruitment process.**

divided by the square of height (kg/m$^2$). Fasting venous blood samples were collected from each participant to routine blood parameters, blood glucose, lipid profile, alanine aminotransferase (ALT), and aspartate transaminase (AST) [22].

A structured questionnaire was administered to each participant to collect information on tobacco and alcohol consumption, as well as physical activity over the past seven days. This questionnaire was developed specifically for our study, drawing upon the World Health Organization's guidelines for controlling and monitoring the tobacco epidemic [23] and previous research on alcohol consumption patterns among young adults in China [24]. According to the frequency of tobacco consumption of the observed subjects, those who smoked more than one cigarette per day for at least six consecutive or cumulative months were defined as smokers. Those who smoked more than four cigarettes per week but less than one cigarette per day were defined as occasional smokers. Those who had never or rarely smoked were categorized as nonsmokers [23]. If the frequency of alcohol consumption of the observed subjects was > 1 drink/month, they were categorized as the drinking group, < 1 drink/month but > 1 drink/year as the occasional drinking group, and < 1 drink/year as the non-drinking group [24]. The total intensity of the participants' physical activities in the past seven days was estimated based on the metabolic equivalents (METs) from the International Physical Activity Questionnaire (IPAQ) short-form survey. To minimize potential recall bias and subjectivity associated with self-reported data, participants were invited to complete the questionnaire together with a family member who could assist in recalling and validating the information provided. Individuals with total metabolic equivalents (METS) ≥ 3000 were categorized into the high-intensity group, those with METS between 600–3000 were in the moderate-intensity group, and the rest were categorized into the low-intensity group [25].

## HP detection

The detection of HP was conducted using the 13C-urea breath test (13C-UBT), a widely recognized non-invasive diagnostic method for HP infection [26–28]. This test measures the isotopic composition of exhaled carbon dioxide after the ingestion of 13C-labeled urea, which is metabolized by the urease enzyme produced by HP. All participants underwent the 13C-UBT on an empty stomach in the morning to ensure standardized testing conditions. The Delta Over Baseline (DOB) value, derived from the isotopic analysis, was used to determine infection status. A DOB value ≥ 4 indicated HP infection (DMHP + group), while a DOB value < 4 indicated no infection (DMHP- group). The DOB threshold of ≥ 4 for determining HP infection was based on established criteria [29,30].

To ensure the reliability of the test results, participants were screened for potential confounding factors. It was confirmed that none of the participants had taken antibiotics, gastric mucosal protectants, or proton pump inhibitors (PPIs) within the preceding month, as these substances can interfere with the urease activity and compromise test accuracy. This protocol aligns with established guidelines for the use of 13C-UBT in clinical and research settings. According to the guideline for primary care of HP infection at the primary level [31], the UBT is considered the gold standard for detecting active infections, as it directly measures urease activity and avoids the risk of false positives associated with serological tests. While serological testing can detect HP antibodies, it cannot distinguish between active and past infections, potentially leading to misclassification of individuals who have recovered from infection.

## 10-year cardiovascular disease risk

The Framingham Risk Score (FRS) was used to assess the 10-year risk of CVD in the study participants. The FRS model included age, gender, HDL-C, TC, systolic blood pressure, and

tobacco consumption status. Although FRS has been used to evaluate the risk of CVD for patients with diabetes mellitus [32,33], it is important to note that it may underestimate CVD risk in diabetic populations due to their inherently higher baseline risk. Alternative tools, such as the UKPDS Risk Engine [34], have been developed for diabetic populations, but the FRS was selected in this study for its widespread acceptance and applicability.

Based on the risk level corresponding to the FRS, the participants were categorized as a high-risk group (>20% 10-year risk), intermediate-risk group (10%-20% 10-year risk), or low-risk group (<10% 10-year risk). These thresholds have been widely used and validated in both general and specific populations, as demonstrated in studies exploring their application in diverse contexts, including biomarker research, diabetic populations, and cardiovascular risk prediction models [32,33,35].

## Statistical analysis

Normally distributed continuous variables were presented as mean ± standard deviation, non-normally distributed continuous variables were described using median and range, and categorical variables were presented as percentages. The Kruskal-Wallis H test was used to assess the statistical differences between DMHP + and DMHP- groups for continuous variables. Statistical differences between groups for categorical variables were obtained by chi-square test. If the count variable has a theoretical number < 10, it is derived using Fisher's exact probability test. The chi-square test was used to calculate the statistical differences for categorical variables between DMHP + and DMHP- groups. Univariate analyses were performed to identify potential factors that affect TC, TG, LDL-C, and HDL-C levels. Multiple logistic regression analysis was used to determine whether HP infection was an independent predictor of lipid metabolism abnormalities in patients with diabetes, and both unadjusted and adjusted models were proposed. Statistical analysis was performed using SPSS (version 25.0), EmpowerStats software (www.empowerstats.com), and R software. A p-value < 0.05 was considered statistically significant.

## Results

### Participant recruitment

A total of 317 participants met the inclusion criteria and were initially included in the study. However, after applying the exclusion criteria, 51 individuals were excluded for the following reasons: 9 individuals refused to measure their height and weight; 23 participants had missing information on the questionnaire; 11 participants were diagnosed with conditions such as tumors; 8 participants had liver-related conditions, including hepatitis, cirrhosis, or schistosomiasis-related liver disease; 4 participants had unreliable laboratory test results. Ultimately, a total of 266 participants were enrolled in the study and formed the final study population.

### Basic characteristics of the study population

The final study population comprised 68 males (25.56%) and 198 females (74.44%). As shown in Table 1, among all patients with diabetes, 90 individuals (33.8%) were infected with HP, including 22 males (24.4%) and 68 females (75.6%), while 176 individuals (66.17%) were not infected, including 46 males (26.1%) and 130 females (73.9%). There were no statistically significant differences between the DMHP + and DMHP- groups in terms of age, sex, RBC, ALT, height, weight, BMI, waist, systolic and diastolic blood pressure, AST, TG, fasting blood glucose, smoking and alcohol consumption (all P values > 0.05). However, there are statistically significant differences in WBC, platelet count, and exercise status between DMHP- and

DMHP+, with p-values < 0.05. As shown in Table 1 (Fig 2), compared with DMHP-, the DMHP+ group had higher TC and LDL- C and lower HDL-C (p-values < 0.05).

## Univariate analysis

We performed univariate analysis to identify possible factors affecting lipid metabolism in DM patients. The results (Table 2) showed that TC levels were elevated by 16.1 mg/dL, LDL-C levels were elevated by 11.4 mg/dL, and HDL-C levels were decreased by 9.0 mg/dL in the DMHP+ group compared to the DMHP− group (p < 0.05). HDL-C levels increased by a mean of 10.2 mg/dL in men with diabetes compared to women (p < 0.05). Age was negatively

**Table 1. General characteristics of patients with diabetes, with and without HP infection.**

| | DMHP- | DMHP+ | Standardize diff. | P-value |
|---|---|---|---|---|
| Number | 176 | 90 | | |
| Age (years, mean ± SD) | 57.3 ± 8.2 | 59.2 ± 9.1 | 0.2 (−0.0, 0.5) | 0.073 |
| Sex (n, %) | | | 0.0 (−0.2, 0.3) | 0.765 |
| Female | 130 (73.9%) | 68 (75.6%) | | |
| Male | 46 (26.1%) | 22 (24.4%) | | |
| Smoking status (n, %) | | | 0.1 (−0.1, 0.4) | 0.734 |
| Nonsmokers | 116 (65.9%) | 60 (66.7%) | | |
| Occasional smoking | 12 (6.8%) | 4 (4.4%) | | |
| Smokers | 48 (27.3%) | 26 (28.9%) | | |
| Drinking state (n, %) | | | 0.1 (−0.2, 0.3) | 0.766 |
| Nondrinkers | 82 (46.6%) | 46 (51.1%) | | |
| Occasional alcohol consumption | 36 (20.5%) | 16 (17.8%) | | |
| Drinkers | 58 (33.0%) | 28 (31.1%) | | |
| Exercise Habits (n, %) | | | 0.4 (0.1, 0.6) | **0.021** |
| Low-intensity group | 124 (71.3%) | 76 (86.4%) | | |
| Medium-intensity group | 38 (21.8%) | 8 (9.1%) | | |
| High-intensity group | 12 (6.9%) | 4 (4.5%) | | |
| RBC (*10^12/L, mean ± SD) | 4.8 ± 0.5 | 4.8 ± 0.4 | 0.0 (−0.2, 0.3) | 0.723 |
| WBC (*10^12/L, mean ± SD) | 6.1 ± 1.7 | 6.6 ± 1.5 | 0.4 (0.1, 0.6) | **0.006** |
| Platelets (*10^12/L, mean ± SD) | 163.5 ± 53.8 | 186.8 ± 63.4 | 0.4 (0.1, 0.7) | **0.002** |
| ALT (U/L, mean ± SD) | 25.7 ± 14.1 | 25.0 ± 17.6 | 0.0 (−0.2, 0.3) | 0.746 |
| Height (cm, mean ± SD) | 161.3 ± 7.7 | 161.4 ± 9.2 | 0.0 (−0.2, 0.3) | 0.912 |
| Weight (Kg, mean ± SD) | 64.2 ± 9.3 | 63.6 ± 9.8 | 0.1 (−0.2, 0.3) | 0.613 |
| BMI (kg/m2, mean ± SD) | 24.6 ± 2.9 | 24.3 ± 2.7 | 0.1 (−0.1, 0.4) | 0.422 |
| Waist (cm, mean ± SD) | 86.7 ± 8.3 | 87.3 ± 8.0 | 0.1 (−0.2, 0.3) | 0.586 |
| SBP (mmHg, mean ± SD) | 131.9 ± 20.3 | 136.2 ± 21.8 | 0.2 (−0.0, 0.5) | 0.111 |
| DBP (mmHg, mean ± SD) | 76.7 ± 11.5 | 75.8 ± 9.6 | 0.1 (−0.2, 0.3) | 0.528 |
| AST (U/L, mean ± SD) | 24.8 ± 11.5 | 24.3 ± 14.4 | 0.0 (−0.2, 0.3) | 0.735 |
| TG (mg/dL, mean ± SD) | 202.4 ± 174.9 | 188.6 ± 174.1 | 0.1 (−0.2, 0.3) | 0.541 |
| TC (mg/dL, mean ± SD) | 183.3 ± 46.3 | 199.3 ± 39.6 | 0.4 (0.1, 0.6) | **0.005** |
| LDL-C (mg/dL, mean ± SD) | 103.9 ± 32.9 | 115.3 ± 32.7 | 0.3 (0.1, 0.6) | **0.008** |
| HDL-C (mg/dL, mean ± SD) | 57.1 ± 12.2 | 48.1 ± 13.5 | 0.7 (0.4, 1.0) | **<0.001** |
| Fasting glucose (mmol/L, mean ± SD) | 8.4 ± 3.1 | 8.7 ± 2.4 | 0.1 (−0.2, 0.3) | 0.566 |

RBC red cell count, WBC white cell count, ALT alanine aminotransferase, AST aspartate transaminase, SBP systolic blood pressure, DBP diastolic blood pressure. Bolding has been applied to all significant p-values (p < 0.05) for improved visibility.

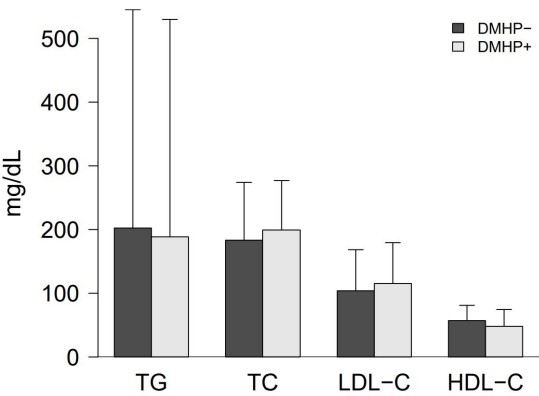

**Fig 2. Differences in lipid metabolism in patients with diabetes, with and without HP infection.**

correlated with TG levels and positively correlated with HDL-C levels (p < 0.05). Alanine aminotransferase (ALT), aspartate transaminase (AST), fasting blood glucose, height, and weight were positively correlated with triglyceride (TG) levels, whereas systolic blood pressure (SBP) was negatively correlated with TG levels (p < 0.05). Fasting blood glucose was positively correlated with TC levels, while weight was negatively correlated with TC levels (p < 0.05). Platelet count was positively correlated with LDL-C levels, whereas ALT, AST, and weight were negatively correlated with LDL-C levels (p < 0.05). Red blood cell count (RBC), white blood cell count (WBC), height, weight, waist circumference, and systolic blood pressure were negatively correlated with HDL-C levels (p < 0.05). HDL-C was lower in drinkers compared with non-drinkers (p < 0.05).

## Relationship between HP infection and lipid metabolism

Factors found by univariate analysis that may affect lipid levels in patients with diabetes were included in multivariate logistic regression to test whether HP infection is an independent risk factor for dyslipidemia in patients with diabetes. As shown in Table 3, in the unadjusted model, TC and LDL-C levels were elevated HDL-C levels were reduced in the DMHP + group compared with the DMHP- group (all P values < 0.05). There was no difference in TG levels between the two groups. In the model adjusted for age and sex (Adjust I) and the model adjusted for all potential confounders (Adjust II), higher TC and LDL-C levels and lower HDL-C levels remained in the DMHP + group than in the DMHP- group (all P-values < 0.05).

In this study, we included variables such as age, sex, smoking status, alcohol consumption, and exercise. Due to data limitations, we were unable to include diabetes duration, glycated hemoglobin (HbA1C), glomerular filtration rate (GFR), or treatment for diabetes and dyslipidemia in the analysis. Specifically, diabetes duration was not recorded in the questionnaire, HbA1C and GFR data were incomplete as they were voluntary tests, and treatment data was not available because the physical examination did not involve medication documentation.

## Cardiovascular disease risk over the next 10 years

As shown in Table 4, there was a statistically significant difference in FRS between DMHP- and DMHP + groups (11.1 ± 3.2 vs. 13.5 ± 3.2, P < 0.001). After stratification according to 10-year CVD risk, the proportions of low, intermediate, and high-risk groups in the DMHP-group were 67.4%, 26.7%, and 5.8%, respectively. In the DMHP + group, the proportions in

**Table 2. Factors potentially influencing lipid metabolism.**

| | Statistics | TG (mg/dL) | TC (mg/dL) | LDL-C (mg/dL) | HDL-C (mg/dL) |
|---|---|---|---|---|---|
| 13C-UBT | | | | | |
| DMHP- | 176 (66.2%) | Reference | Reference | Reference | Reference |
| DMHP+ | 90 (33.8%) | −13.8 (−58.2, 30.5) 0.541 | 16.1 (4.8, 27.3) **0.005** | 11.4 (3.1, 19.7) 0.008 | −9.0 (−12.2, −5.8) **<0.001** |
| Sex (n,%) | | | | | |
| Female | 198 (74.4%) | Reference | Reference | Reference | Reference |
| Male | 68 (25.6%) | −41.5 (−89.4, 6.4) 0.090 | 6.2 (−6.1, 18.5) 0.325 | −1.6 (−10.7, 7.5) 0.731 | 10.2 (6.7, 13.6) **<0.001** |
| Smoking status (n, %) | | | | | |
| Nonsmokers | 176 (66.2%) | Reference | Reference | Reference | Reference |
| Occasional smoking | 16 (6.0%) | −10.3 (−96.3, 75.8) 0.815 | −27.8 (−50.6, −5.1) **0.017** | −15.6 (−32.4, 1.2) 0.070 | −5.2 (−11.8, 1.5) 0.128 |
| Smokers | 74 (27.8%) | 107.4 (61.7, 153.1) **<0.001** | −5.4 (−17.4, 6.7) 0.383 | −9.1 (−18.1, −0.2) **0.045** | −7.0 (−10.5, −3.5) **<0.001** |
| Drinking state (n, %) | | | | | |
| Nondrinkers | 128 (48.1%) | Reference | Reference | Reference | Reference |
| Occasional alcohol consumption | 52 (19.5%) | 6.0 (−47.3, 59.3) 0.826 | −13.1 (−27.5, 1.3) 0.075 | −6.6 (−17.2, 4.1) 0.229 | −5.6 (−9.9, −1.4) **0.010** |
| Drinkers | 86 (32.3%) | 124.0 (78.8, 169.2) **<0.001** | 1.4 (−10.8, 13.5) 0.827 | −7.7 (−16.8, 1.3) 0.094 | −4.0 (−7.6, −0.4) **0.029** |
| Exercise Habits (n, %) | | | | | |
| Low-intensity group | 200 (76.3%) | Reference | Reference | Reference | Reference |
| Medium-intensity group | 46 (17.6%) | 28.3 (−27.2, 83.8) 0.319 | 24.4 (10.5, 38.4) **<0.001** | 15.8 (5.3, 26.2) **0.003** | 5.0 (0.7, 9.3) **0.022** |
| High-intensity group | 16 (6.1%) | 121.2 (33.0, 209.4) **0.008** | 17.9 (−4.2, 40.1) 0.114 | 5.7 (−10.8, 22.3) 0.498 | −2.1 (−8.9, 4.6) 0.536 |
| Age (years, mean ± SD) | 57.9 ± 8.5 | −5.1 (−7.5, −2.7) **<0.001** | 0.1 (−0.6, 0.7) 0.805 | 0.2 (−0.3, 0.6) 0.513 | 0.2 (0.1, 0.4) **0.010** |
| RBC (*10¹²/L, mean ± SD) | 4.8 ± 0.5 | 7.7 (−38.2, 53.5) 0.744 | −4.1 (−15.8, 7.7) 0.499 | 7.0 (−1.6, 15.7) 0.113 | −7.1 (−10.5, −3.7) **<0.001** |
| WBC (*10¹²/L, mean ± SD) | 6.3 ± 1.6 | −0.5 (−13.3, 12.3) 0.937 | −0.1 (−3.3, 3.2) 0.974 | 0.9 (−1.5, 3.3) 0.464 | −1.6 (−2.5, −0.6) **0.001** |
| Platelets (*10¹²/L, mean ± SD) | 171.4 ± 58.1 | −0.3 (−0.7, 0.0) 0.088 | 0.1 (−0.0, 0.2) 0.209 | 0.1 (0.0, 0.1) **0.034** | −0.0 (−0.0, 0.0) 0.155 |
| ALT (U/L, mean ± SD) | 25.5 ± 15.4 | 2.9 (1.6, 4.2) **<0.001** | −0.3 (−0.7, 0.0) 0.076 | −0.4 (−0.7, −0.2) **0.001** | −0.0 (−0.1, 0.1) 0.620 |
| AST (U/L, mean ± SD) | 24.7 ± 12.5 | 3.0 (1.4, 4.6) **<0.001** | −0.3 (−0.7, 0.1) 0.183 | −0.5 (−0.8, −0.2) **0.001** | 0.1 (−0.1, 0.2) 0.393 |
| Fasting glucose (mmol/L, mean ± SD) | 8.5 ± 2.9 | 19.3 (12.4, 26.2) **<0.001** | 3.9 (2.1, 5.7) **<0.001** | 1.2 (−0.2, 2.5) 0.098 | 0.1 (−0.5, 0.7) 0.735 |
| Height (cm, mean ± SD) | 161.4 ± 8.3 | 2.6 (0.1, 5.2) **0.041** | −0.6 (−1.3, 0.0) 0.068 | −0.3 (−0.8, 0.2) 0.265 | −0.5 (−0.6, −0.3) **<0.001** |
| Weight (Kg, mean ± SD) | 64.0 ± 9.5 | 2.4 (0.2, 4.6) **0.035** | −0.6 (−1.2, −0.1) **0.029** | −0.4 (−0.8, −0.0) **0.045** | −0.3 (−0.5, −0.2) **<0.001** |
| BMI (kg/m2, mean ± SD) | 24.5 ± 2.8 | 3.2 (−4.2, 10.6) 0.397 | −1.3 (−3.2, 0.6) 0.190 | −1.2 (−2.6, 0.2) 0.100 | −0.3 (−0.8, 0.3) 0.359 |
| SBP (mmHg, mean ± SD) | 133.3 ± 20.9 | −1.5 (−2.5, −0.5) **0.003** | −0.0 (−0.3, 0.2) 0.861 | 0.0 (−0.2, 0.2) 0.837 | 0.1 (0.0, 0.2) **0.040** |
| DBP (mmHg, mean ± SD) | 76.4 ± 10.9 | −0.8 (−2.7, 1.1) 0.419 | −0.4 (−0.9, 0.1) 0.146 | −0.1 (−0.5, 0.2) 0.440 | −0.0 (−0.2, 0.1) 0.758 |
| Waist (cm, mean ± SD) | 86.9 ± 8.2 | 1.8 (−0.8, 4.4) 0.173 | −0.3 (−1.0, 0.3) 0.352 | −0.2 (−0.7, 0.3) 0.359 | −0.2 (−0.4, −0.0) **0.016** |

Data in table: β (95%CI) P-value/ OR (95%CI) P-value. Bolding has been applied to all significant p-values (p < 0.05) for improved visibility. β coefficients are presented for continuous variables, and Odds Ratios (ORs) for categorical variables (underlined). The term "Reference" indicates the baseline category for comparison in categorical variables.

these three groups were 34.9%, 34.9%, and 30.2%, respectively. The difference in risk composition between the two groups was statistically significant (p < 0.001, Fig 3).

## Discussion

This study found that the DMHP+ group had higher TC and LDL-C levels than DMHP-, while HDL-C levels were lower than DMHP-. However, there was no statistically significant difference in TG levels between the DMHP- and DMHP+ groups. Even after adjusting for all factors affecting lipid metabolism in the logistic regression model, infection with HP was an independent risk factor for lower HDL-C and higher TC and LDL-C levels in patients with diabetes. Our study showed that patients with diabetes with HP infection had higher FRS and

**Table 3. Relationship between HP infection and lipid metabolism abnormalities in patients with diabetes.**

|  | Non-adjusted (β, 95% CI, P) | Adjust I (β, 95% CI, P) | Adjust II (β, 95% CI, P) |
|---|---|---|---|
| **TG (mg/dL)** |  |  |  |
| DMHP- | Reference | Reference | Reference |
| DMHP+ | −13.8 (−58.2, 30.5) 0.5413 | −4.7 (−48.0, 38.7) 0.833 | 4.0 (−38.9, 47.0) 0.854 |
| **TC (mg/dL)** |  |  |  |
| DMHP- | Reference | Reference | Reference |
| DMHP+ | 16.1 (4.8, 27.3) **0.0054** | 16.3 (5.0, 27.7) **0.005** | 14.3 (3.0, 25.5) **0.014** |
| **LDL-C (mg/dL)** |  |  |  |
| DMHP- | Reference | Reference | Reference |
| DMHP+ | 11.4 (3.1, 19.7) **0.0078** | 11.1 (2.7, 19.5) **0.010** | 10.4 (1.8, 18.9) **0.018** |
| **HDL-C (mg/dL)** |  |  |  |
| DMHP- | Reference | Reference | Reference |
| DMHP+ | −9.0 (−12.2, −5.8) **< 0.001** | −9.3 (−12.3, −6.3) **< 0.001** | −10.0 (−13.2, −6.7) **< 0.001** |

Non-adjusted: No adjustment for any factors. The term "Reference" indicates the baseline category for comparison in categorical variables. Bolding has been applied to all significant p-values (p < 0.05) for improved visibility. Adjust I: Adjusted for age and gender. Adjusted II adjusted for: Gender; Age; Red blood cell count; Decreased white blood cell count; Platelet count; ALT; Height; Weight; BMI; Waist circumference; Systolic blood pressure; Diastolic blood pressure; Smoking and alcohol consumption status; Exercise status; AST; Fasting blood glucose.

**Table 4. Differences in FRS and cardiovascular disease risk between the DMHP- and DMHP + groups.**

|  | DMHP- | DMHP+ | Standardize diff. | P-value |
|---|---|---|---|---|
| N | 176 | 90 |  |  |
| FRS | 11.1 ± 3.2 | 13.5 ± 3.2 | 0.8 (0.5, 1.0) | **<0.001** |
| Risk grouping |  |  | 0.8 (0.5, 1.1) | **<0.001** |
| Low risk | 116 (67.4%) | 30 (34.9%) |  |  |
| Medium risk | 46 (26.7%) | 30 (34.9%) |  |  |
| High risk | 10 (5.8%) | 26 (30.2%) |  |  |

Results in table: Mean + SD/ N (%). Bolding has been applied to all significant p-values (p < 0.05) for improved visibility.

a higher proportion of people at risk of CVD. We observed that HP may be a risk factor for increased risk of dyslipidemia and cardiovascular disease in patients with diabetes, consistent with previous findings [36,37].

HP infection may contribute to dyslipidemia in patients with diabetes through several intertwined mechanisms involving inflammation and metabolic dysregulation. Chronic HP infection induces systemic inflammation by increasing the production of pro-inflammatory cytokines such as interleukin-6 (IL-6) and tumor necrosis factor-alpha (TNF-α) [38]. These cytokines can exacerbate insulin resistance by interfering with insulin signaling pathways [39–41]. Heightened insulin resistance is a hallmark of type 2 diabetes and is closely linked to alterations in lipid metabolism, including increased LDL-C and decreased HDL-C levels [42]. Additionally, HP-associated inflammation may impair hepatic function, reducing the liver's capacity to metabolize cholesterol effectively [43,44]. This impairment can lead to the accumulation of LDL-C and a decrease in HDL-C. Furthermore, HP infection may disrupt gut microbiota composition, which plays a crucial role in lipid absorption and metabolism [45–47]. Dysbiosis induced by HP could therefore contribute to dyslipidemia by altering bile acid metabolism and influencing lipid regulatory genes [21,36,48].

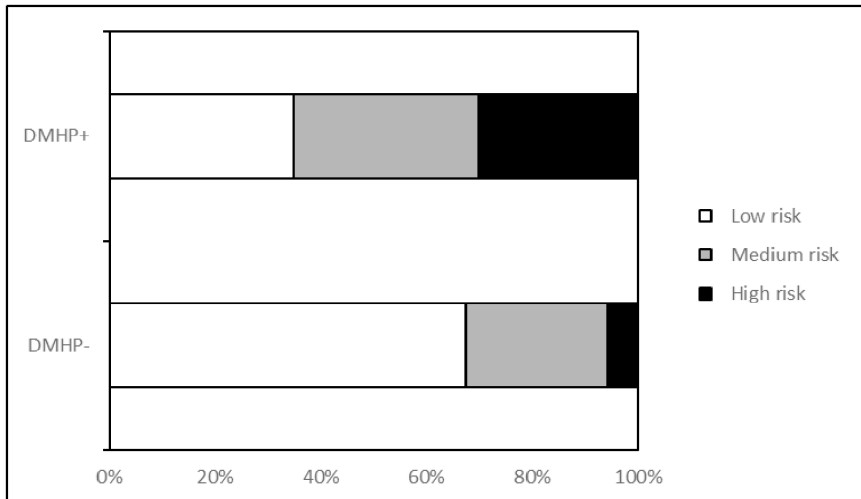

**Fig 3. Differences in cardiovascular disease risk between diabetic patients with and without HP infection.**

Although these mechanisms are biologically plausible, it is important to emphasize that we did not investigate any of these pathways directly in the present study. Consequently, the proposed explanations for how HP may affect lipid profiles remain speculative based on the existing literature. We recommend that future studies explore these pathways in depth—potentially through mechanistic and longitudinal research designs—in order to validate the hypotheses regarding HP-induced inflammatory, hepatic, and gut microbiota changes that might contribute to dyslipidemia in patients with diabetes.

Despite the changes observed in LDL-C and HDL-C, our study found no significant difference in triglyceride (TG) levels between the DMHP− and DMHP+ groups. This finding may be explained by the different metabolic pathways governing cholesterol and triglycerides. While cholesterol metabolism is heavily influenced by hepatic function and inflammatory processes, triglyceride levels are more directly affected by factors such as insulin resistance, obesity, and dietary intake of fats and carbohydrates [49–51]. In patients with diabetes, insulin resistance tends to promote hypertriglyceridemia due to increased very-low-density lipoprotein (VLDL) production [52,53]. However, the effect of HP on TG levels may be less pronounced because the bacterium's impact on lipid metabolism primarily affects cholesterol synthesis and transport rather than triglyceride synthesis. Additionally, medications commonly used in diabetes management, such as metformin or statins, could modulate TG levels independently of HP infection [54,55]. The lack of significant difference in TG levels suggests that HP infection may have a selective effect on lipid fractions, warranting further investigation into the specific pathways involved.

There are inconsistencies in the literature regarding how the severity of HP infection affects its association with dyslipidemia. Some studies indicate that the presence of more virulent HP strains, such as those expressing cytotoxin-associated gene A (CagA), is associated with greater alterations in lipid profiles [56,57]. These strains may induce stronger inflammatory responses, leading to more significant disruptions in lipid metabolism. Conversely, other studies have not found a clear relationship between HP infection severity and dyslipidemia [58,59]. Factors contributing to these inconsistencies may include differences in study populations, variations in bacterial strains, genetic predispositions of hosts, and environmental influences such as diet and lifestyle. Additionally, methodological differences, such as the diagnostic criteria for assessing infection severity, could lead to variable results. Our study did not evaluate the severity or

strain-specific characteristics of HP infection, which may limit our understanding of its impact on lipid metabolism. Future research should focus on delineating how different HP strains and infection intensities influence dyslipidemia in patients with diabetes.

Our study has many strengths. First, it confirms that comorbid HP infection is a risk factor for increased risk of dyslipidemia and cardiovascular disease in patients with diabetes. Second, the study adequately adjusted for potential confounders affecting lipid metabolism and demonstrated that HP infection was independently associated with dyslipidemia in patients with diabetes. Rooting out HP is significant for patients with diabetes. In addition, the study demonstrated the possibility of regulating lipid metabolism in patients with diabetes by controlling HP infection in the digestive tract, providing a new idea for future lipid-lowering therapy.

However, there are limitations to this study. As a cross-sectional investigation, it can only demonstrate the correlation between HP infection and abnormalities of lipid metabolism, and cannot clarify the causation between them. This research is insufficient to determine whether eradicating HP would improve lipid metabolism and reduce the risk of cardiovascular disease in patients with diabetes. Therefore, future studies should compare changes in lipid levels and cardiovascular disease risk before and after HP eradication treatment in individuals with diabetes. The higher number of female participants in this study may be related to the fact that there were more female diabetes checkup clients during the study period. Although there was no statistical difference in gender composition between the DMHP- and DMHP+ groups in this study, future studies should still consider increasing the proportion of male observers.

A notable limitation of this study is the absence of certain key variables in our dataset, such as diabetes duration, glycated hemoglobin (HbA1C), glomerular filtration rate (GFR), and treatment regimens for diabetes and dyslipidemia. These variables are critical for understanding the multifaceted nature of diabetes and its associated risks. For example, diabetes duration and HbA1C levels are well-established indicators of long-term glycemic control and may influence susceptibility to HP infection and related complications. Similarly, GFR reflects renal function, which can act as a confounding factor in studies involving diabetes populations. The lack of data on these variables was primarily due to the nature of our dataset, which was derived from routine physical examinations. HbA1C and GFR tests were voluntary, leading to incomplete data, while diabetes duration and treatment regimens were not documented in the questionnaire. As a result, these variables could not be included in our analysis, potentially introducing residual confounding. Despite these limitations, we adjusted for other relevant confounders, including age, sex, and exercise, to mitigate potential biases. Future research should aim to incorporate these critical variables to provide a more comprehensive understanding of the interplay between diabetes and HP infection. Collecting detailed data on diabetes management, glycemic control, and renal function would allow for more robust analyses and could yield valuable insights into the mechanisms underlying these associations.

Additionally, the methods used to assess physical activity in this study, specifically the METs (Metabolic Equivalent of Task) and IPAQ (International Physical Activity Questionnaire) short-form surveys, have inherent limitations that should be acknowledged. While these tools are widely used and validated for estimating physical activity levels, they rely on self-reported data, which is subject to recall bias and social desirability bias. Participants may overestimate or underestimate their physical activity levels, leading to potential inaccuracies in the data. Furthermore, the IPAQ short-form survey provides a general measure of physical activity but lacks detailed information on specific types of activity (e.g., aerobic versus resistance exercise) or intensity levels, which may have differential effects on lipid metabolism and cardiovascular risk. The use of these tools, while practical in large-scale studies, may limit the precision of our findings regarding the role of physical activity as a confounder or modifier in the relationship between HP infection and lipid abnormalities.

To minimize potential recall bias and subjectivity associated with self-reported data, participants were invited to complete the questionnaire together with a family member who could assist in recalling and validating the information provided. This approach aimed to enhance the accuracy of the self-reported physical activity data by providing an additional layer of verification. Despite these efforts, the limitations of self-reported data remain, and future studies should consider incorporating objective measures of physical activity, such as accelerometers or pedometers, to validate self-reported data and provide more granular insights into the impact of physical activity on these associations.

## Conclusion

Concurrent HP infection is associated with dyslipidemia in individuals with diabetes. Compared to patients with diabetes without HP infection, those with concurrent infection exhibit elevated FRS and a potentially increased 10-year risk of cardiovascular diseases. However, the cross-sectional nature of this study limits the ability to infer causation, and further longitudinal studies are warranted to confirm these findings.

## Supporting information

**S1 Data.  The data.**
(XLSX)

**S2 Data.  Zhujieforhpdm.**
(XLSX)

## Author contributions

**Conceptualization:** Yuexi Li.

**Data curation:** Yuexi Li.

**Formal analysis:** Xiaoqin Liu.

**Funding acquisition:** Xiaoqin Liu.

**Investigation:** Qing Li.

**Methodology:** Qing Li.

**Project administration:** Peng Zhou.

**Resources:** Peng Zhou.

**Software:** Qian Chen.

**Supervision:** Qian Chen.

**Validation:** Bolan Jiang.

**Visualization:** Bolan Jiang.

**Writing – original draft:** Yuexi Li, Taiju Zhu.

**Writing – review & editing:** Taiju Zhu.

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
