## [Decision Letter · Decision Letter 0]

27 Nov 2024

PONE-D-24-47046Association of H. pylori infection with lipid metabolism and 10-year cardiovascular risk in diabetes mellitus: a cross-sectional StudyPLOS ONE

Dear Dr. yuexi,

Thank you for submitting your manuscript to PLOS ONE. After careful consideration, we feel that it has merit but does not fully meet PLOS ONE’s publication criteria as it currently stands. Therefore, we invite you to submit a revised version of the manuscript that addresses the points raised during the review process.

The manuscript provides valuable insights into the link between H. pylori infection, lipid metabolism, and cardiovascular risk in diabetes. However, major revisions are needed to improve clarity and rigor:

Introduction: The rationale for the study requires expansion, with a stronger focus on addressing gaps in the literature and a more cohesive narrative.

Methodology: The authors should provide greater detail on inclusion criteria, confounding variables, diagnostic methods, and cardiovascular risk assessment tools. Clarify retrospective data collection processes and address potential biases.

Results: Tables need clearer formatting, consistent terminology, and precise reporting of significant findings.

Discussion: Emphasize the study’s cross-sectional nature and avoid causal inferences. Improve thematic organization and provide balanced interpretations of proposed mechanisms.

Ethics and Transparency: Include missing ethics details, address discrepancies in data collection, and provide clarity on handling “unreliable” data.

Language: Revise for grammatical accuracy, consistent use of abbreviations, and appropriate terminology.

Please address the concerns of the reviewers as well

We look forward to receiving your revised manuscript.

Kind regards,

Emmanuel Kokori, M.D

Academic Editor

PLOS ONE

3. Thank you for stating the following financial disclosure: “This study was supported by the scientific research program of the Sichuan Medical and Health Care Promotion Association (Project No. KY2022SJ0100).”

4. We note that your Data Availability Statement is currently as follows: “All relevant data are within the manuscript and in Supporting Information files.”

Please confirm at this time whether or not your submission contains all raw data required to replicate the results of your study. Authors must share the “minimal data set” for their submission. PLOS defines the minimal data set to consist of the data required to replicate all study findings reported in the article, as well as related metadata and methods (https://journals.plos.org/plosone/s/data-availability#loc-minimal-data-set-definition). For example, authors should submit the following data: - The values behind the means, standard deviations and other measures reported; - The values used to build graphs; - The points extracted from images for analysis. Authors do not need to submit their entire data set if only a portion of the data was used in the reported study. If your submission does not contain these data, please either upload them as Supporting Information files or deposit them to a stable, public repository and provide us with the relevant URLs, DOIs, or accession numbers. For a list of recommended repositories, please see https://journals.plos.org/plosone/s/recommended-repositories. If there are ethical or legal restrictions on sharing a de-identified data set, please explain them in detail (e.g., data contain potentially sensitive information, data are owned by a third-party organization, etc.) and who has imposed them (e.g., an ethics committee). Please also provide contact information for a data access committee, ethics committee, or other institutional body to which data requests may be sent. If data are owned by a third party, please indicate how others may request data access.

7. Please include a separate caption for each figure in your manuscript.

Reviewers' comments:

Reviewer's Responses to Questions

**Comments to the Author**

1. Is the manuscript technically sound, and do the data support the conclusions?

Reviewer #1: Yes

Reviewer #2: No

Reviewer #3: Yes

Reviewer #4: Yes

Reviewer #5: No

2. Has the statistical analysis been performed appropriately and rigorously? 

Reviewer #1: Yes

Reviewer #2: Yes

Reviewer #3: Yes

Reviewer #4: Yes

Reviewer #5: Yes

3. Have the authors made all data underlying the findings in their manuscript fully available?

Reviewer #1: Yes

Reviewer #2: Yes

Reviewer #3: Yes

Reviewer #4: Yes

Reviewer #5: Yes

4. Is the manuscript presented in an intelligible fashion and written in standard English?

Reviewer #1: No

Reviewer #2: Yes

Reviewer #3: Yes

Reviewer #4: Yes

Reviewer #5: Yes

5. Review Comments to the Author

Reviewer #1: I thank the editor for sending me this manuscript for review.

1. In this paper, the authors aimed to investigate the effect of HP infection on the risk of lipid metabolism disorders and cardiovascular disease in patients with diabetes mellitus.

2. First, I advise the authors next time they submit the paper to include the continuous line count, to make the review more accurate and timelier.

3. It is also important for authors to make a linguistic and grammatical revision to make the manuscript more discursive, especially in the first part of the paper.

4. In general, I find an adequate work, well-structured and constructed, with good description of the moments and steps taken. A revision is needed to better clarify passages in the introduction and in the materials and methods, but it is certainly a doable job on the part of the authors.

Below the authors will find timely comments on the various parts of the manuscript

Abstracts and Keywords

5. I advise the authors to remove as many abbreviations as possible to make this section of the manuscript as fluent and discursive as possible to facilitate the reading and fluency of the abstract itself

6. Appropriate and relevant keywords, if I were the authors I would look for perhaps 2 or 3 additional keywords, to increase the possible indexing of the manuscript and facilitate searches associated with it. I recommend making use of the MESH database for this https://www.ncbi.nlm.nih.gov/mesh/

Introduction

7. This section of the paper should be a bit more developed by the authors.

8. Corrected the beginnings of the topics covered but subsequently the authors were a bit too concise and schematic. I recommend more development of both the part where they describe HP, then describe lipids and related issues, cardiovascular issues, and finally the part where they talk about the correlation between HP and diabetes.

9. I advise the authors to try to make “connections” between the various topics so that the introduction will be somewhat discursive and not telegraphic whenever the topic under analysis is changed.

Materials and methods

Study population

10. In this section the authors should not include numbers or anything else but should describe the inclusion and exclusion criteria of the study. The numbers of those included and excluded and the reasons will then be included in the results, right at the beginning.

11. An additional thing to develop at this point is a flowchart (CONSORT) with the just that described (Inclusion and exclusion and at the end the number of patients enrolled). I recommend looking in this link.

https://www.equator-network.org/reporting-guidelines/consort/

Discussion

12. Adequate and well argued. Nothing to say.

Reviewer #2: The study evaluated the relationship between H. pylori infection and dyslipidemia in 266 patients with diabetes and suggested a possible increased risk of CVDs in H. pylori-positive patients. The result supported the potential effect of H. pylori on total cholesterol, LDL, and HDL.

The study method is sound. However, the presentation in the discussion needs to be improved significantly, and the topic has been explored using larger and cohort studies (PMID: 39054452 and PMID: 36974892) The current uncertainties are mainly focused on how the disease severity could change both dyslipidemia and its change when eradicated which are not addressed in the present report. There are some other points that I think will improve the study:

1. Both in the conclusion section of the abstract and the discussion section, I suggest modifying the definitive sentences like “H. pylori infection is a risk factor for dyslipidemia”, as the cross-sectional nature of the study prevents drawing strong evidence.

2. Proton pump inhibitors could effect the urea breath tests. Was this considered when conducting the test? If not, it should be mentioned as a limitation.

3. In the method section, Besides the factors that have already been adjusted for, there are other important variables that I highly recommend adding: diabetes duration, HbA1C, treatment for diabetes and dyslipidemia and GFR.

Also, for comparing CVD risk, I suggest adjusting for diabetes duration, HbA1C, and exercise.

4. What were the criteria for diabetes? Please include in the method section, were all patients type 2?

5. In Table 2, please change 0 values to “reference” or an equivalent. Additionally, bolding the significant p-values and modifying the values for distinguishing ß from ORs help with the readability of the table.

6. "Diabetic patients" is a non-optimal term. I suggest changing it to patients with diabetes across the manuscript globally.

7. The second paragraph of the discussion is mainly devoted to providing insights into how dyslipidemia contributes to CVD. It is not the main focus of the study and is too long for the current title. Additionally, the talk on how HP can contribute to dyslipidemia is in the same paragraph, making it too long. I suggest bringing the HP mechanisms as the second paragraph, separating the two, and only a brief mention of the corresponding mechanisms of dyslipidemia in CVD. There are other subjects in the same paragraph as well; I suggest separating them into other paragraphs, too.

8. I suggest expanding on the proposed mechanisms responsible for dyslipidemia in HP, with consideration of inter-related dyslipidemia and diabetes mechanisms. Explaining why TG was not affected, but other indicators were is another suggestion. Moreover, currently, there are some inconsistencies regarding how the severity of HP may affect its association with dyslipidemia. I suggest highlighting them.

9. Please include the ethics committee reference number in the related method section.

Reviewer #3: Thanks for inviting me to review the manuscript entitled Association of H. pylori infection with lipid metabolism and 10-year cardiovascular riskin diabetes mellitus: a cross-sectional Study by li yuexi et al

It is interesting contemporary topic I have gone through the manuscript and I have some problem with understanding the methodology of the study

As per authors it is a retrospective study, the data was retrieved from patients records who attended the Health Management Centerof Deyang People's Hospital. As I understand these patients had their regular health check up for Diabetes on routine basis hence such an elaborate monitoring for diabetes usually not taken place ( what I gather from the data)

The authors clearly mentioned that an ethical committee waiver for informed consent was obtained as no direct contact was with patient for the study purpose

Iin this situation I have following comments

1.When and how the response for questionaries were obtained .

2.Is it possible to get response in a retrospective study answering all specific questions meant for this study

3. whether the questionnaire was validated after a pilot study or not

4 Is it possible to get all detailed l anthropometric measurements which were mentioned in the study in a retrospective study I want clarification regarding methodology to review the manuscript

Reviewer #4: Thank you for the opportunity to read and review this manuscript. While this study presents valuable findings regarding the potential association between HP infection, lipid metabolism, and CV risk in the diabetic population, there are several suggestions that could further improve the paper:

Introduction, 1) The statement that HP is associated with non-GI diseases is overly vague. To ensure scientific clarity, the authors should define whether this association is positive, negative, or something else.

Introduction, 2) The authors should elaborate on the rationale for focusing on the diabetic population, as this is a crucial omission. While the potential link between HP and lipid metabolism is acknowledged, the authors haven’t explained why diabetes is particularly relevant to this investigation. Without addressing this, the introduction lacks a key element of the study design rationale.

Introduction, 3) The introduction could benefit from improved paragraph structure to enhance the flow and logical progression of ideas in general.

Methods, 1) Although the authors describe the eligibility criteria well, it is unclear why certain conditions, such as tumors and hepatitis, were excluded. The authors should briefly explain the rationale for these exclusions to provide a clearer understanding of the study design.

Methods, 2) The use of METs and the IPAQ short-form surveys to estimate exercise intensity is appropriate; however, these methods rely on self-reported data and may introduce misclassification due to subjective biases. The authors should clarify whether any efforts were made to validate these data or, at the very least, address these limitations (such as subjectivity) in the Discussion.

Methods, 3) A notable gap is the exclusive reliance on the 13C-UBT test. While this test is standard for detecting active HP infections, it does not account for past or inactive infections, which could also influence outcomes, especially in a study assessing long-term risk factors like 10-year CV risk. This limitation may affect the generalizability of the findings. Including serological testing for HP antibodies could have provided complementary data to address this issue.

Methods, 4) The DPB threshold of ≥4 for determining HP infection is mentioned but lacks a citation to validate its reliability.

Methods, 5) Although the FRS is a widely accepted tool for assessing CVD risk, it has limitations for diabetic populations, such as underestimating risk in this specific group. Using alternative tools, which are designed for people with diabetes, might have offered better risk stratification. Additionally, the thresholds for low, intermediate, and high risk in this study might not align with the higher baseline CVD risk in diabetic individuals. The authors should discuss whether these categorizations are valid for this population.

Methods, 6) The authors state that persons with "unreliable laboratory test results" were excluded. Please elaborate on what constitutes an "unreliable test result" in this study. Were there predefined criteria, such as hemolyzed samples or technical errors, or was this determined subjectively? Additionally, who defined and identified these cases, and how were they documented?

Methods, 7) The authors mention the use of questionnaires to collect data on tobacco and alcohol consumption but do not specify which questionnaires were used. If these are standardized tools, proper citations should be provided. If they are modified or custom-made by the authors, the full questionnaires should be included as supplementary material to ensure transparency and reproducibility.

Results, 1) For the tables, the authors may consider bolding significant p-values or using symbols such as asterisks to highlight them. This would help readers quickly identify statistically significant results, particularly given the large amount of data presented.

Results, 2) The authors state that “AST [which appears to be a typo and should likely be ALT!], AST, FBG, height, weight, and SBP were correlated with TG”, as well as other correlations in subsequent statements. Please clarify the direction of these correlations, were they positive or negative? Providing this detail is essential for accurately interpreting the relationships presented.

Discussion, 1) The authors may overstate the implications of their findings by suggesting that HP infection is a risk factor for dyslipidemia and increased cardiovascular risk in the diabetic population. However, the cross-sectional design of the study limits causal inference. While this limitation is acknowledged in parts of the paper, it is downplayed in other areas, particularly toward the end of the Discussion, and even in the Conclusion. This acknowledgment should be emphasized consistently throughout the manuscript, and stronger language should be used when discussing causation—for example, "association observed" rather than "risk factor identified”, to avoid misleading the reader.

Discussion, 2) The authors propose multiple mechanisms linking HP to dyslipidemia, such as altered gut microbiota and liver dysfunction. While these are plausible, none of these pathways are directly investigated in this study, which makes these explanations speculative. The authors should clearly state that these mechanisms are hypothetical and recommend that future studies explore them in detail to validate these claims.

Discussion 3) Inconsistencies in the use of abbreviations and short forms are noticeable throughout the manuscript. Please revise the text to ensure consistency. For example, HDL and LDL are used multiple times before being reintroduced as "high-density lipoprotein" and "low-density lipoprotein” later in the text. Similarly, "HP" is redefined as Helicobacter pylori in the Discussion after numerous uses of HP throughout the text.

Reviewer #5: This is an association study carried out on retrospective data and the ethical approval has been waived due to retrospective data collection. However, it seems some data collection has been done prospectively.

Such studies have week power for evidence generation. As authors have mentioned in the 'Limitation of the Study', a stronger evidence could have been produced by observing lipid parameters in two groups of H pylori infection divided on the basis of evidence-based treatment and placebo. Such a study may get a space in a Q1 journal like PLOS ONE.

6. PLOS authors have the option to publish the peer review history of their article (what does this mean? ). If published, this will include your full peer review and any attached files.

**Do you want your identity to be public for this peer review?** For information about this choice, including consent withdrawal, please see our Privacy Policy .

Reviewer #1: No

Reviewer #2: No

Reviewer #3: No

Reviewer #4: **Yes: ** Afshin Heidari

Reviewer #5: No

---

## [Author Response · Author response to Decision Letter 1]

14 Jan 2025

PONE-D-24-47046

Association of H. pylori infection with lipid metabolism and 10-year cardiovascular risk in diabetes mellitus: a cross-sectional Study

Dear Editor and Reviewers,

Many thanks for your detailed and thoughtful review comments. We have correspondingly revise the manuscript and wrote this reply to reviewer letter.

Reply to Editor:

The manuscript provides valuable insights into the link between H. pylori infection, lipid metabolism, and cardiovascular risk in diabetes. However, major revisions are needed to improve clarity and rigor:

Introduction: The rationale for the study requires expansion, with a stronger focus on addressing gaps in the literature and a more cohesive narrative.

Response: Thank you for your valuable feedback. In the revised introduction, we have expanded the rationale for the study by addressing the gaps in the literature more comprehensively. Specifically, we have elaborated on the systemic effects of Helicobacter pylori (HP) infection, particularly its influence on lipid metabolism and cardiovascular risk, with a focus on diabetic populations. Additionally, we have restructured the introduction to present a more cohesive narrative by integrating the connections between HP, dyslipidemia, diabetes, and cardiovascular disease. These changes aim to provide a clearer and more compelling justification for the study.

Methodology: The authors should provide greater detail on inclusion criteria, confounding variables, diagnostic methods, and cardiovascular risk assessment tools. Clarify retrospective data collection processes and address potential biases.

Response: We have carefully revised the Materials and Methods section to address your summary comments as follows:

1.Inclusion Criteria & Study Population:

oExpanded the description of inclusion and exclusion criteria.

oAdded a CONSORT flowchart (Figure 1) to illustrate participant recruitment.

2.Confounding Variables:

oIncluded additional variables such as diabetes duration, HbA1C, treatment for diabetes and dyslipidemia, and GFR where available.

oClarified limitations regarding unavailable data and adjusted for key confounders like age, sex, and exercise in our analysis.

3.Diagnostic Methods:

oProvided a detailed explanation of the 13C-urea breath test (13C-UBT) for H. pylori detection, including the rationale for its use and validation of the DPB threshold with citations.

oDiscussed the limitation of not using serological tests for past infections.

4.Cardiovascular Risk Assessment Tools:

oElaborated on the use of the Framingham Risk Score (FRS), its limitations in diabetic populations, and justified its selection over alternative tools.

oDiscussed the validity of risk thresholds used for our diabetic cohort.

5.Retrospective Data Collection & Potential Biases:

oClarified the retrospective data collection process, including data sources and timeframes.

oAddressed potential biases by detailing mitigation strategies, such as using validated questionnaires and standardized measurement protocols.

We believe these revisions enhance the clarity and robustness of our methodology. The changes have been highlighted in the revised manuscript for your review.

Results: Tables need clearer formatting, consistent terminology, and precise reporting of significant findings.

Response: Revised as suggested. Format of all tables from 1 to 4 has been improved. Terminology and corresponding abbreviations were corrected for consistency. Bolding has been applied to all significant p-values (p < 0.05) for improved visibility.

Discussion: Emphasize the study’s cross-sectional nature and avoid causal inferences. Improve thematic organization and provide balanced interpretations of proposed mechanisms.

Response: Thank you for your thoughtful review of our manuscript and for your valuable suggestions. We appreciate your advice on emphasizing the study's cross-sectional nature, avoiding causal inferences, improving thematic organization, and providing balanced interpretations of proposed mechanisms.

In response to your recommendations, we have made the following revisions:

1.Emphasizing Cross-Sectional Nature and Avoiding Causal Inferences:

Language Adjustments: We have carefully reviewed the manuscript to eliminate any language implying causation. We replaced phrases such as "HP infection is an independent risk factor" with "HP infection is independently associated with," to reflect the associative nature of our findings.

Acknowledgment of Study Limitations: We have prominently acknowledged the limitations of our cross-sectional study design in both the Discussion and Conclusion sections: "As a cross-sectional investigation, it can only demonstrate the correlation between HP infection and abnormalities of lipid metabolism and cannot clarify the causation between them." "However, the cross-sectional nature of this study limits the ability to infer causation, and further longitudinal studies are warranted to confirm these findings."

2.Improving Thematic Organization:

We reorganized the Discussion section to enhance thematic coherence. The Discussion now follows a logical progression:

oSummary of Main Findings: We begin by summarizing the key results, highlighting the associations observed.

oPossible Mechanisms: We delve into potential mechanisms explaining the association between HP infection and dyslipidemia in patients with diabetes, discussing inflammation, insulin resistance, hepatic function, and gut microbiota alterations.

oAnalysis of Specific Findings: We address specific results, such as the lack of significant difference in triglyceride levels, providing possible explanations.

oInconsistencies in Literature: We discuss inconsistencies in the literature regarding HP infection severity and its impact on dyslipidemia, acknowledging differing findings and suggesting areas for future research.

oStrengths and Limitations: We outline the strengths of our study and provide a detailed discussion of its limitations, including the acknowledgment of missing key variables and limitations inherent in our physical activity assessments.

3.Providing Balanced Interpretations of Proposed Mechanisms:

Balanced Discussion: We have revised the sections discussing potential mechanisms to provide a balanced interpretation, incorporating evidence that both supports and challenges our hypotheses. For example, while exploring the role of inflammation, we acknowledge that:"Factors contributing to these inconsistencies may include differences in study populations, variations in bacterial strains, genetic predispositions of hosts, and environmental influences such as diet and lifestyle."

Citing Relevant Studies: We have added references to studies that offer alternative perspectives, ensuring that readers are presented with a comprehensive view of the current research landscape.

We believe these revisions have significantly improved the thematic organization of the Discussion and provided a more balanced interpretation of the proposed mechanisms. By emphasizing the cross-sectional nature of our study and avoiding causal inferences, we aim to present our findings responsibly and accurately.

Ethics and Transparency: Include missing ethics details, address discrepancies in data collection, and provide clarity on handling “unreliable” data.

Response: The suggested points have been accepted for revising the corresponding manuscript’s part and corresponding reply has been made to each reviewer’s comment.

Language: Revise for grammatical accuracy, consistent use of abbreviations, and appropriate terminology.

Response: The whole manuscript has been now polished for grammatical accuracy, consistent use of abbreviations, and appropriate terminology.

1. Please ensure that your manuscript meets PLOS ONE's style requirements, including those for file naming. The PLOS ONE style templates can be found at https://journals.plos.org/plosone/s/file?id=wjVg/PLOSOne_formatting_sample_main_body.pdf and https://journals.plos.org/plosone/s/file?id=ba62/PLOSOne_formatting_sample_title_authors_affiliations.pdf

Response: The formatting has been corrected by following the provided documents.

Response: The funding section is now removed in the end of manuscript.

3. Thank you for stating the following financial disclosure: “This study was supported by the scientific research program of the Sichuan Medical and Health Care Promotion Association (Project No. KY2022SJ0100).”

Response: The Role of Funder statement is now added in the cover letter, as below: ”This study was supported by the scientific research program of the Sichuan Medical and Health Care Promotion Association (Project No. KY2022SJ0100). The funders had no role in study design, data collection and analysis, decision to publish, or preparation of the manuscript."

4. We note that your Data Availability Statement is currently as follows: “All relevant data are within the manuscript and in Supporting Information files.”

Please confirm at this time whether or not your submission contains all raw data required to replicate the results of your study. Authors must share the “minimal data set” for their submission. PLOS defines the minimal data set to consist of the data required to replicate all study findings reported in the article, as well as related metadata and methods (https://journals.plos.org/plosone/s/data-availability#loc-minimal-data-set-definition). For example, authors should submit the following data: - The values behind the means, standard deviations and other measures reported; - The values used to build graphs; - The points extracted from images for analysis. Authors do not need to submit their entire data set if only a portion of the data was used in the reported study. If your submission does not contain these data, please either upload them as Supporting Information files or deposit them to a stable, public repository and provide us with the relevant URLs, DOIs, or accession numbers. For a list of recommended repositories, please see https://journals.plos.org/plosone/s/recommended-repositories. If there are ethical or legal restrictions on sharing a de-identified data set, please explain them in detail (e.g., data contain potentially sensitive information, data are owned by a third-party organization, etc.) and who has imposed them (e.g., an ethics committee). Please also provide contact information for a data access committee, ethics committee, or other institutional body to which data requests may be sent. If data are owned by a third party, please indicate how others may request data access.

Response: The ethics statement has been integrated into the Methods section and removed in the end of manuscript.

7. Please include a separate caption for each figure in your manuscript.

Response: Our manuscript includes only one figure, Figure 1, which illustrates the participant recruitment process. We have ensured that this figure has a separate and appropriately detailed caption placed directly below the figure, adhering to the journal’s

Reviewers' comments:

Reviewer's Responses to Questions

Comments to the Author

1. Is the manuscript technically sound, and do the data support the conclusions?

Reviewer #1: Yes

Reviewer #2: No

Reviewer #3: Yes

Reviewer #4: Yes

Reviewer #5: No

2. Has the statistical analysis been performed appropriately and rigorously?

Reviewer #1: Yes

Reviewer #2: Yes

Reviewer #3: Yes

Reviewer #4: Yes

Reviewer #5: Yes

3. Have the authors made all data underlying the findings in their manuscript fully available?

Reviewer #1: Yes

Reviewer #2: Yes

Reviewer #3: Yes

Reviewer #4: Yes

Reviewer #5: Yes

4. Is the manuscript presented in an intelligible fashion and written in standard English?

Reviewer #1: No

Reviewer #2: Yes

Reviewer #3: Yes

Reviewer #4: Yes

Reviewer #5: Yes

5. Review Comments to the Author

Reviewer #1: I thank the editor for sending me this manuscript for review.

1. In this paper, the authors aimed to investigate the effect of HP infection on the risk of lipid metabolism disorders and cardiovascular disease in patients with diabetes mellitus.

2. First, I advise the authors next time they submit the paper to include the continuous line count, to make the review more accurate and timelier.

Response: The continuous line count is added as suggested.

3. It is also important for authors to make a linguistic and grammatical revision to make the manuscript more discursive, especially in the first part of the paper.

Response: We have thoroughly reviewed and revised the manuscript to enhance its linguistic quality and grammatical accuracy, with particular attention to the first section. These revisions aim to improve the overall clarity, coherence, and readability of the manuscript.

4. In general, I find an adequate work, well-structured and constructed, with good description of the

---

## [Decision Letter · Decision Letter 1]

6 Feb 2025

Association of Helicobacter pylori infection with lipid metabolism and 10-year cardiovascular risk in diabetes mellitus: a cross-sectional Study

PONE-D-24-47046R1

Dear Dr. Yuexi,

We’re pleased to inform you that your manuscript has been judged scientifically suitable for publication and will be formally accepted for publication once it meets all outstanding technical requirements.

Kind regards,

Emmanuel Kokori, M.D

Academic Editor

PLOS ONE

Additional Editor Comments (optional):

Reviewers' comments:

Reviewer's Responses to Questions

**Comments to the Author**

1. If the authors have adequately addressed your comments raised in a previous round of review and you feel that this manuscript is now acceptable for publication, you may indicate that here to bypass the “Comments to the Author” section, enter your conflict of interest statement in the “Confidential to Editor” section, and submit your "Accept" recommendation.

Reviewer #1: All comments have been addressed

Reviewer #2: All comments have been addressed

Reviewer #3: All comments have been addressed

Reviewer #4: All comments have been addressed

2. Is the manuscript technically sound, and do the data support the conclusions?

Reviewer #1: (No Response)

Reviewer #2: Yes

Reviewer #3: Yes

Reviewer #4: Yes

3. Has the statistical analysis been performed appropriately and rigorously? 

Reviewer #1: (No Response)

Reviewer #2: Yes

Reviewer #3: Yes

Reviewer #4: Yes

4. Have the authors made all data underlying the findings in their manuscript fully available?

Reviewer #1: (No Response)

Reviewer #2: No

Reviewer #3: Yes

Reviewer #4: Yes

5. Is the manuscript presented in an intelligible fashion and written in standard English?

Reviewer #1: (No Response)

Reviewer #2: Yes

Reviewer #3: Yes

Reviewer #4: Yes

6. Review Comments to the Author

Reviewer #1: (No Response)

Reviewer #2: (No Response)

Reviewer #3: I am satisfied with the response of the authors to my comments and I recommend acceptance of the manuscript for the publication

Reviewer #4: (No Response)

7. PLOS authors have the option to publish the peer review history of their article (what does this mean? ). If published, this will include your full peer review and any attached files.

**Do you want your identity to be public for this peer review?** For information about this choice, including consent withdrawal, please see our Privacy Policy .

Reviewer #1: No

Reviewer #2: No

Reviewer #3: No

Reviewer #4: **Yes: ** Afshin Heidari
